# Molecular driving force of a small molecule-induced protein disorder-order transition
Cesar Mendoza-Martinez [1,2,5], Arun A. Gupta[1,3,5], Salomé Llabrés [1,4,5], Paul N. Barlow[1] & Julien Michel [1] ✉

The selectivity and affinity of numerous protein–protein interactions depends upon the folding of intrinsically disordered regions (IDRs) that accompanies complexation. Here we investigate how folding-on-binding of a protein IDR by small molecules is facilitated by synergestic exploitation of interactions with a folded protein region. To this end, the molecular driving forces that underpin ordering of the N-terminal intrinsically disordered 'lid' region of the oncoprotein MDM2 by the small molecule AM-7209 were elucidated by a combination of molecular dynamics simulations, calorimetry and NMR measurements. Strikingly, mutations of lid residues distant from the ligand-binding site modulate potency by up to three orders of magnitude. A key requirement for conversion of this IDR into an ordered motif is collective stabilisation of a network of non-polar contacts between a chlorophenyl moiety of AM-7209 and the lid residue I19 to overcome conformational entropy loss associated with folding of the IDR. Our findings underscore the crucial role that protein IDRs can play in drug-resistance mechanisms and expand strategies available to medicinal chemists for ligand optimisation endeavours.

Many crucial biochemical processes rely on intricate networks of protein–protein interactions (PPIs). It is widely acknowledged that almost half of the proteome in eukaryotes consists of proteins containing intrinsically disordered regions (IDRs)[1]. These IDRs are characterised as contiguous segments of 20 or more amino acid residues that remain unstructured under native conditions[2,3]. Notably IDRs play a pivotal role in numerous PPIs[4] and in the formation of biomolecular condensates[5,6]. Frequently an IDR undergoes a disorder-to-order transition upon binding to another protein, resulting in the formation of a low-affinity, high-selectivity protein–protein complex[7–10], but there are also known examples of IDRs that remain disordered in a bound protein–protein complex[11,12].

It is well documented that PPIs play a crucial role in the pathogenesis of diverse and complex diseases, such as cancer, diabetes and neurodegenerative disorders. Unsurprisingly modulation of PPIs with small molecules is an appealing drug discovery strategy[13]. However targeting PPIs with small-molecule drugs has proven to be a formidable challenge, with only a limited number of such inhibitors successfully becoming approved drugs over the past quarter-century.

Different examples of small molecules have been reported to form disordered bound complexes with target IDRs, often with relatively weak binding affinity[14–21]. The lack of evidence of ordering of an IDR on direct binding by a small molecule may be a consequence of the smaller interaction contact surface between a protein-small molecule complex versus a protein–protein complex. A deeper understanding of how to design small molecules that emulate PPI recognition mechanisms, such as the folding of a protein IDR upon binding, may unlock new strategies for developing drug candidates and new possibilities for effective interventions in complex disease processes.

This report focuses on unravelling the molecular driving forces of a disorder-to-order transition of a protein IDR induced by a small molecule. Targeting the p53/MDM2 protein–protein interaction is a well-established anticancer strategy and several small-molecule inhibitors have reached late-stage clinical trials. However the emergence of drug resistance due to mutations in p53 or MDM2 has underscored the need for developing next-generation inhibitors[22]. Most p53/MDM2 inhibitors bind to the N-terminal domain of MDM2, which consists of a core structured region and an IDR

[1]EaStCHEM School of Chemistry, University of Edinburgh, Edinburgh, UK. [2]Present address: Drug Discovery Unit, University of Dundee, Dundee, UK. [3]Present address: Nuffield Department of Medicine, Center for Immuno-Oncology, University of Oxford, Oxford, UK. [4]Present address: Faculty of Pharmacy and Food Science, Universitat de Barcelona, Barcelona, Spain. [5]These authors contributed equally: Cesar Mendoza-Martinez, Arun A. Gupta, Salomé Llabrés. ✉e-mail: julien.michel@ed.ac.uk

**Fig. 1 | Ligand-specific modulation of conformational preferences of the lid IDR of the MDM2 N-terminal domain. A** In apo MDM2 the lid region is unstructured and exchanges on millisecond timescale between 'open' and 'closed' conformational states[53]. This regulates access to the p53-binding site. **B** Binding of AM-7209 is accompanied by ordering of the MDM2 lid region into a helix-turn-strand motif.

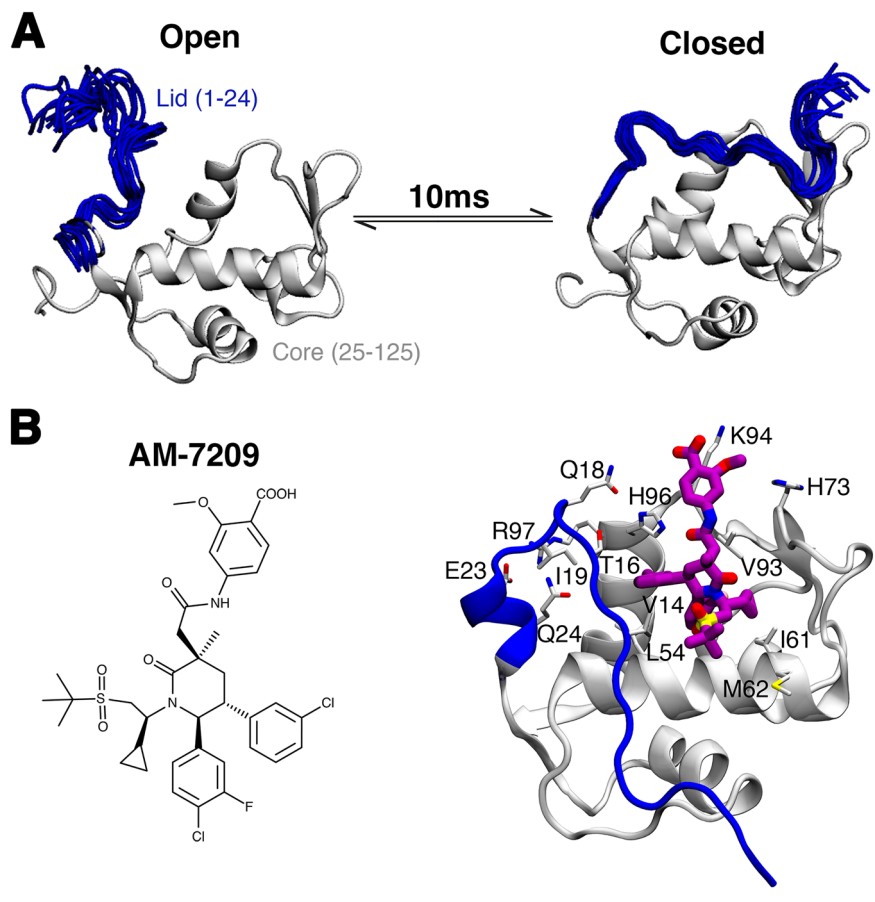

referred to as a 'lid' (Fig. 1A)[23]. AM-7209, a clinical candidate compound belonging to a family of piperidinone molecules, uniquely orders the N-terminal 'lid' of MDM2 upon binding (Fig. 1B)[24]. Previous work from our group has elucidated why AM-7209 selectively bind to the ordered-lid conformation of MDM2, whereas other compound classes bind without ordering the lid[25,26]. Thus the MDM2 lid IDR provides a model system to investigate synergistic exploitation of structured and disordered protein region features for small molecule ligand optimisation programs.

Here we achieve a deep understanding of the molecular driving forces that underpin AM-7209:lid recognition by performing a combination of molecular dynamics (MD) simulations, isothermal titration calorimetry (ITC) experiments and nuclear magnetic resonance (NMR) measurements[27–35]. Significantly, we identify a crucial amino residue in the intrinsically disordered lid of MDM2 that modulates AM-7209 affinity by up to three orders of magnitude. These insights provide valuable design principles for targeting future IDRs flanking structured protein regions with small molecules and highlight the potential role of IDRs in drug-resistance mechanisms.

## Results and discussion
### Computational design of mutants that modulate disorder-to-order transition propensities
Previous experimental and computational work has demonstrated that the N-terminal lid region of MDM2 adopts a helix-turn-strand motif in the presence of several piperidinone ligands[25,36]. This folding-on-binding event is driven by a complex balancing of changes in protein–ligand polar and non-polar contacts together with changes in protein, ligand and solvent entropy. We hypothesised that monitoring the helical stability of the ordered helix-turn-strand motif would provide a convenient metric to identify lid mutations that would significantly interfere with the folding-on-binding mechanism. A panel of single-site and double-site MDM2 mutants was prepared for in silico analyses to probe the contributions to the stabilisation

of this motif of various lid-to-protein and lid-to-ligand contacts (Fig. 1B). Mutations to a glycine were attempted at several locations, as this residue shows significant differences in backbone conformational preferences over other amino acid residues and would be expected to increase the inherent structural disorder of the lid. Each mutant underwent two independent 0.5 µs equilibrium MD simulations and the stability of the helical motif observed at the base of the lid was assessed by post-processing of the computed trajectories.

Single substitutions closer to the N terminus (residues 14–18) did not significantly disrupt the helical motif, although some mutations (V14G, V14D and T16G) increased positional fluctuations of the extended segment (Fig. 2B, C). Modifications of I19 that decreased the size of the side chain (I19G, I19A) significantly decreased helical propensity and significantly increased positional fluctuations. This was explained by the loss of non-polar contacts with residues T16, L54 and Y100. Interestingly, replacement of I19 by E19 was tolerated as the helical motif was stabilised by compensating interactions between E19 and R97.

Previous work suggested that the salt-bridge between E23 and R97 is critical to lock the lid in an ordered conformation[25]. However the present MD simulations suggested that a broad range of mutations at E23 (E23G, E23L and E23Q) cause only moderate loss of helicity and did not significantly increase lid flexibility. The double mutant E23R97:R23E97 was simulated to test the effect of swapping the residues involved in salt-bridge formation. This causes a moderate decrease in the stability of the helical motifs. Other double mutants that included I19G were simulated to evaluate potential additive effects (I19G:E23G, V14G:I19G, T16G:I19G). These double mutants exhibited helical stability comparable to that observed in I19G, suggesting that mutations of I19 were key to destabilising the ordered lid state (Fig. S20).

We stress that ~µs timescale equilibrium MD simulations are too short to reliably characterise the MDM2 lid folding-on-binding process and their analysis was restricted here to the identification of key lid residues that could contribute to the stabilisation of the ordered lid state. Further detailed

**Fig. 2 | Designing mutants with molecular dynamics simulations. A** Key lid-residues selected for mutation studies. **B** Helix propensity of lid residues 21–25, for a panel of single and double mutants, calculated from MD simulations. Error bars denote $\pm 1\sigma_E$. **C** Calculated root-mean squared fluctuations for MDM2 residues 6–28 for the same panel of mutants.

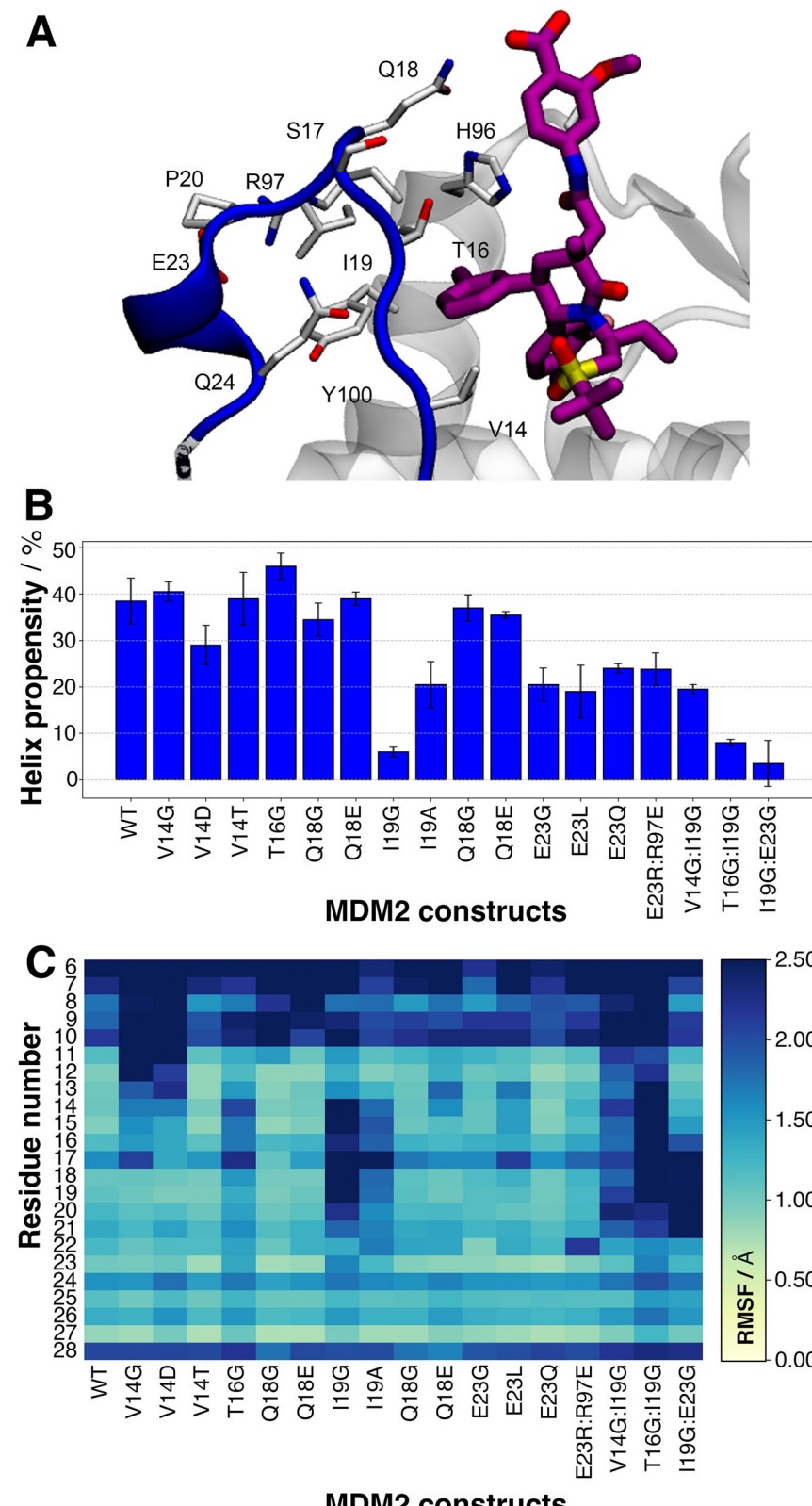

analyses of the folding-on-binding mechanism were therefore pursued with biophysical experiments detailed below.

### Calorimetry measurements identify mutations that modulate AM-7209 affinity by three orders of magnitude

A panel of eight mutants, selected on the basis of the MD simulations, was subjected to biophysical measurements to further investigate the

relationships between predicted lid helix propensity, lid flexibility and ligand binding energetics (Figs. S1–S10). The compound Nutlin-3a, which does not order the intrinsically disordered lid of wild-type (WT) MDM2[36], served as a control (Fig. 3A). As anticipated, the ITC-derived thermodynamic signature of Nutlin-3a binding to MDM2 remained similar for all constructs, irrespective of the lid mutations (Fig. 3B). This suggests that any interactions of Nutlin-3a with the lid, if present, are energetically

**Fig. 3 | Low and high affinity MDM2 variants have a distinct thermodynamic signature of binding.**
**A** X-ray structures of the Nutlin-3a-bound (PDB id 4J3E) and the AM7209 (PDB id 4WT2) bound to the MDM2 protein. **B** ITC-measured thermodynamic signature for the binding of Nutlin-3a to the panel of MDM2 mutants. Green: enthalpy of binding; red: entropy of binding; blue: free energy of binding. **C** Same data for the binding of AM-7209 to the panel of MDM2 mutants. **D** ITC-derived dissociation constants for both ligands. *:' direct titration, ': competitive titration. Uncertainties are the standard deviation of the mean from triplicate measurements.

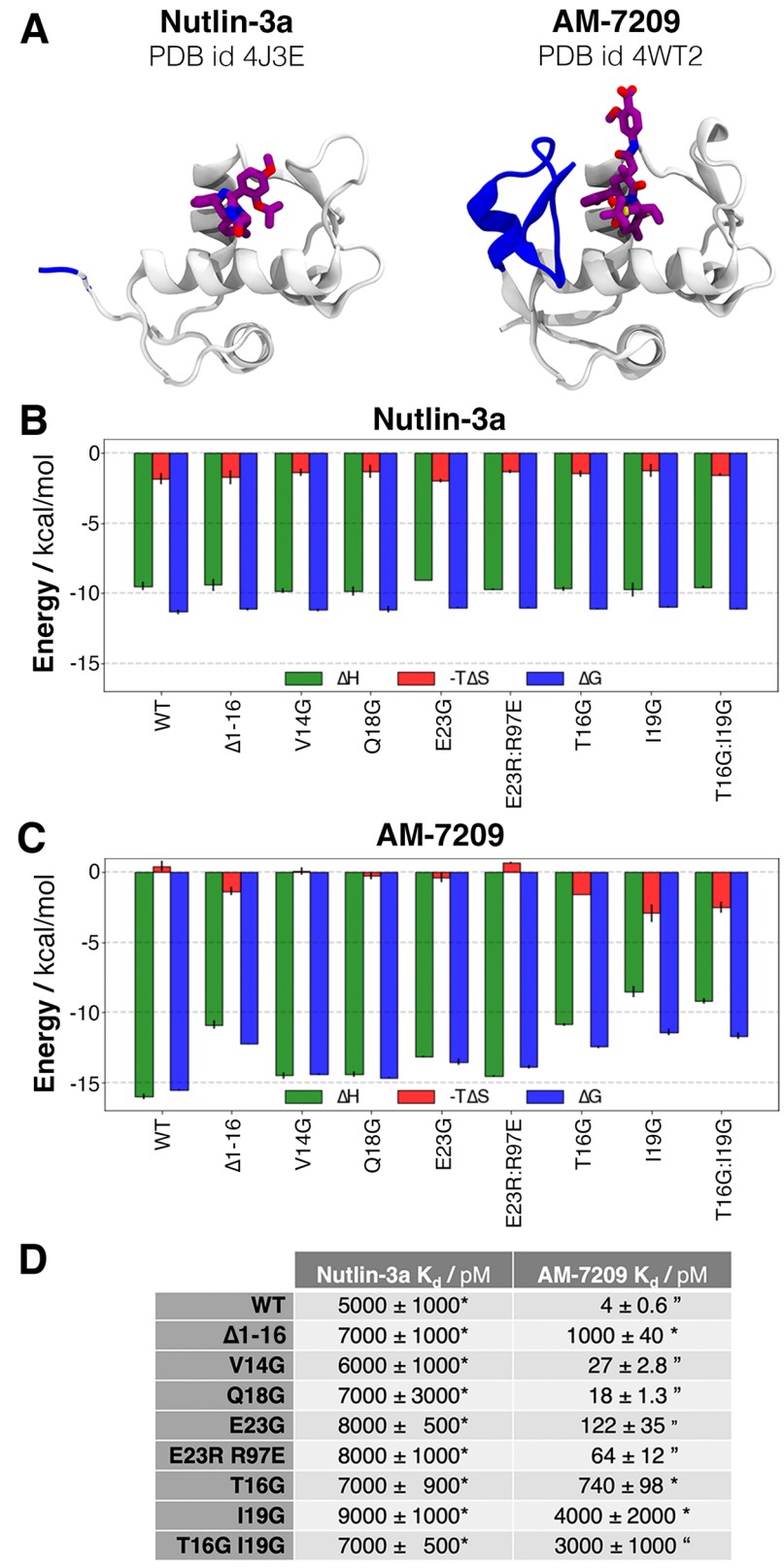

| | Nutlin-3a $K_d$ / pM | AM-7209 $K_d$ / pM |
|---|---|---|
| **WT** | 5000 ± 1000* | 4 ± 0.6 " |
| **Δ1-16** | 7000 ± 1000* | 1000 ± 40 * |
| **V14G** | 6000 ± 1000* | 27 ± 2.8 " |
| **Q18G** | 7000 ± 3000* | 18 ± 1.3 " |
| **E23G** | 8000 ± 500* | 122 ± 35 " |
| **E23R R97E** | 8000 ± 1000* | 64 ± 12 " |
| **T16G** | 7000 ± 900* | 740 ± 98 * |
| **I19G** | 9000 ± 1000* | 4000 ± 2000 * |
| **T16G I19G** | 7000 ± 500* | 3000 ± 1000 " |

inconsequential. This aligns with the notion that Nutlin-3a does not induce ordering of the MDM2 lid region upon binding.

In contrast the binding thermodynamic signature of AM-7209 was markedly influenced by mutations in the lid region (Fig. 3C). Generally, weaker binding mutants exhibited a more negative entropy of binding and a more positive enthalpy of binding. Mutations predicted to have little-to-no

effect on helix stability in the lid region (V14G, Q18G, E23G and E23R:R97E) showed entropies of binding similar to WT. Mutations predicted to disrupt helix formation (I19G, T16G:I19G) exhibited entropies of binding similar to the lid-truncated construct. The T16G mutant behaved intermediately between the two other classes of mutants. This supports the idea that binding of AM-7209 to weaker mutants is entropically favoured

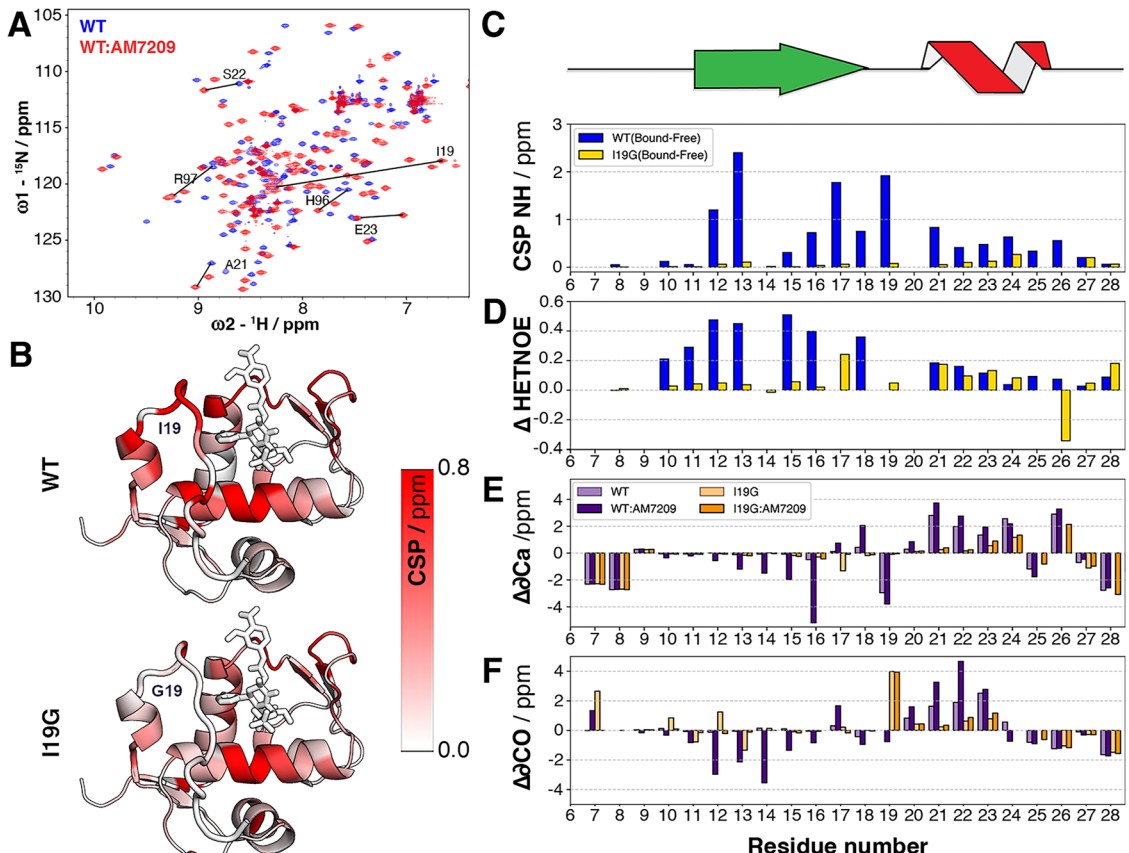

**Fig. 4 | NMR analysis upon titration of AM-7209 into samples of WT MDM2 and I19G mutant. A** Overlays of a region of the $^1H,^{15}N$ HSQC spectra for WT MDM2, with selected lid-residue cross peaks annotated, before titration (blue) and after (red) titration with AM-7029. **B** Projection of combined $^1H,^{15}N$ CSP values for WT and I19G on a representative 3D structure of the WT MDM2:AM-7209 complex. **C** Combined $^1H,^{15}N$ CSPs for MDM2 residues 6–28 induced by AM-7209 addition for WT (blue) and I19G MDM2 (yellow). The location within the AM-7209-stabilised lid conformation of extended (green) and helical (red) regions is depicted above the plot. **D** AM-7209-induced differences in heteronuclear ($^1H$-$^{15}N$) (Het) NOE values for the amides of MDM2 residues 6–28 (ΔHetNOE = HetNOEbound–HetNOEfree). **E** Secondary 13Cα chemical shifts for WT (light purple), WT plus AM-7209 (purple), I19G (light orange), I19G plus AM-7209 (orange). **F** Same as (**E**) but for secondary 13CO chemical shifts.

due to the absence of lid ordering but enthalpically disfavoured because of a reduced number of contacts between the ligand and the lid.

Remarkably, analysis of the ITC-derived binding constants (Fig. 3D) reveals that the affinity of AM-7209 for I19G is decreased by 1000-fold compared to WT, comparable with the affinity measured for the lid-truncated construct. Conversely, mutations of residues that directly contact AM-7209 have a modest effect (V14G, Q18G) or intermediate effect (T16G) on binding constants. The significant effect of the T16G mutation on AM-7209 affinity suggests that the MD simulations may have underestimated the structural perturbation of the lid IDR caused by this mutation. The double mutant T16G:I19G binds AM-7209 with similar affinity to the single mutant I19G, suggesting that formation of the helical motif in the lid to place T16 in close proximity to the ligand necessitates stabilising interactions from I19. The importance of the E23-R97 salt bridge for lid ordering was probed with the E23G mutation. The resultant moderate loss of AM-7209 affinity is consistent with the moderate loss of helical propensity in MD simulations. Interestingly, swapping of salt-bridging residues in the E23R:R97E double mutant does not maintain WT binding affinity. This is consistent with MD simulations indicating a decreased propensity for formation of the R23-E97 salt bridge versus the native E23-R97 salt bridge.

## NMR measurements reveal distinct conformational preferences between wild-type MDM2 and a designed mutant

The I19G mutant was further characterised by preparation of isotopically enriched MDM2 (aa. 6–125) in WT and I19G forms for NMR experiments (Figs. S11 and S12). Comparison of the HSQC spectra of WT and I19G in

apo forms showed large chemical perturbations were limited to residues in the immediate vicinity of residue 19 (Figs. S13 and S17). Significant perturbations in amide ($^1H,^{15}N$) chemical shifts were apparent throughout the protein, including for several lid residues, upon titration (up to a ligand:-protein molar ratio of 1.2:1) of AM-7209 into 50 μM WT MDM2 (Figs. 4A and S14). When the same concentrations of AM-7209 were titrated into 50 μM I19G significant chemical shifts perturbations were observed throughout the protein with the crucial exception of lid residues (Fig. S15). Displaying the combined $^1H,^{15}N$ CSP values on a 3D structure of MDM2, confirms that the large perturbations observed in the lid region of WT MDM2 are absent in the case of the I19G mutant (Fig. 4B, C). Figures S16 and S17 show data for the entire protein sequences.

Heteronuclear ($^1H$-$^{15}N$) NOE values for lid residues 10–17 showed a significant increase following AM-7209 titration into WT MDM2, suggesting decreased mobility on the ps-ns timescale, consistent with ordering of this region (Fig. 4D). Conversely, negligible or small increases in heteronuclear NOE values were observed for the equivalent lid residues 1–16 following AM-7209 titration into the I19G mutant. However small hetNOE and R2 values increases that are comparable to WT were observed for lid residues 19–25 (Fig. 4D, see Figs. S18 and S19 for data for the entire protein sequences). This suggests that internal dynamics of residues 19–25 in I19G is slowed somewhat by AM-7209 binding, even in the absence of large chemical shift changes. In WT MDM2, residues 12–16 showed a trend towards negative secondary 13Ca chemical shifts upon AM-7209 addition, while residues 20–23 showed a trend towards positive 13Ca chemical shifts (Fig. 4E, light purple and purple), consistent with increased β-strand and α-

helical propensity, respectively. These trends were not observed when AM-7209 was titrated into a sample of I19G (Fig. 4E, light orange and orange). Upon addition of AM-7209 to WT MDM2 a trend towards negative secondary 13CO chemical shift values was observed for residues 12–16 as opposed to the positive values observed for residues 20–23, which is again consistent with increased β-strand and α-helical propensity, respectively (Fig. 4F, light purple and purple). No such trends were observed when AM-7209 was added to the I19G mutant (Fig. 4F, light orange and orange).

Taken together these results demonstrate that substitution of MDM2 Ile19 with Gly does not significantly change the nature of the conformational ensemble of the MDM2 lid region in the unbound state. Rather the Ile19 to Gly substitution in the I19G mutant destabilises the ordered helical-turn-extended motif that accompanies binding of AM-7209 in WT.

### Free energy calculations identify the structural requirements for ordering of the MDM2 lid by AM-7209

The earlier MD simulations were crucial to identify the key I19G mutation for experimental characterisation of the lid folding-on-binding mechanism. However µs-time scale equilibrium MD simulations cannot robustly capture the full energy landscape connecting ordered and disordered lid states. We thus undertook extensive free energy surfaces (FES) calculations of the MDM2 lid region in the absence and presence of AM-7209 using our previously described aMD/US/vFEP methodology to complement the interpretation of the calorimetric and NMR measurements (see Figs. S21–S26 and Supplementary Materials 'Methods' paragraph)[25,26,37]. The calculations suggest that the lid region of unliganded WT predominantly adopts a partially disordered conformation that occludes the ligand-binding site (label C1 Fig. 5A). They further suggest that the disordered lid in unliganded MDM2 can also transiently adopt an alternative position that gives partial access to the ligand-binding site (label C2, Fig. 5A). Full 'opening' of the lid is energetically disfavoured (label C3, Fig. 5A).

After AM-7209 binding, the dominant lid conformational state appears to shift to an ordered helical and partially extended motif (label C4, Fig. 5B). An ordered helical and fully extended motif (label C5, Fig. 5B), consistent with X-ray crystallography data, is slightly less energetically favoured. Such a fully extended lid conformation can be stabilised in X-ray diffracted structures via crystal packing. Motif formation requires repositioning of the lid away from core helix α2 to avoid steric clashes with the ligand. A closed disordered motif (label C6, Fig. 5B) with the lid positioned above the ligand is also energetically accessible according to these calculations.

The calculations suggest that in unliganded I19G the energetically dominant lid state is disordered and occludes the ligand binding site (label C8, Fig. 5C). As in unliganded WT a partially open, partially helical motif is energetically feasible (label C9, Fig. 5C). Overall, the energy landscape of the unliganded lid region appears broadly similar for WT and I19G, in agreement with the NMR chemical shift and relaxation measurements (Figs. S13 and S17–S19). This was corroborated by a comparison of CSP values computed with ShiftX2 from the apo ensembles and the NMR measured CSPs that indicated a lack of significant perturbations in chemical shifts for residues away from position 19 (Fig. S27). Upon docking in AM-7209, lid conformations that wrap over the ligand become energetically disfavoured (label C12, Fig. 5D). The lid conformational preferences shift towards adoption of a partially open and collapsed state (label C11, Fig. 5D). Comparison of the states sampled at the lowest values of the WT:AM-7209 and I19G:AM-7209 FES (label C4 Fig. 5B vs label C11 Fig. 5D) shows that in the I19G mutant the lid remains largely disordered. The base of the lid (residues 20–25) does not form a helical motif but rather collapses on itself. These findings are consistent with the small increases in NMR HetNOE and R2 relaxation rate measurements observed in the lid region between the bound and free form of the I19G mutant (Fig. S19).

A secondary structure analysis of the full conformational ensembles predicted by the above FES indicates that binding of AM-7209 to WT significantly increases helicity of residues 21–25 (Fig. 5E, first and second rows). By contrast binding of AM-7209 to I19G is associated with only a small increase in helical propensity for residues 21–25 (Fig. 5E, third and fourth rows). As reported previously, the DSSP algorithm does not detect the strand component of the helix-turn-strand motif in the ordered MDM2 lid IDR (Fig. 5E, second row) around residues 12–16[25]. Nevertheless, adoption of an extended conformation for lid residues 12–16 is apparent from visualisation of the FES (Fig. 5B label C5) and consistent with the increased β-strand propensity for residues 12–16 inferred from secondary chemical shift changes (Fig. 4E, F). The helix propensity around residues 10–17 observed in Fig. 5E is associated with transient helices seen in closed lid conformations that wrap above the ligand (e.g. Fig. 5D label C12).

Additionally, we investigated changes in hydrophobic solvent accessible surface area (SASA) between free and bound forms of WT and I19G to indirectly assess the contribution of changes in solvent entropy to the energetics of the folding-on-binding process (Table S4). We found that both WT and I19G show a similar reduction in protein hydrophobic SASA (decreases of $9.0 \pm 1.0\%$ and $7.5 \pm 0.8\%$ respectively), whereas AM-7209 shows a slightly lower reduction in SASA upon binding I19G rather than WT (decreases of $67.2 \pm 1.0\%$ and $73.0 \pm 0.4\%$ respectively). This suggests that differential solvation of WT and I19G doesn't make a significant contribution to the folding-on-binding process. We additionally approximated changes in conformational entropy of the protein between free and bound forms by post-processing the computed ensembles (see 'Methods'). We find that binding of AM-7209 incurs a conformational entropy loss for WT that is greater than for I19G by ca. 1 kcal/mol (Fig. S28). The greater entropy loss arises from contributions of residues in the lid region. In addition, when we calculated the loss of translational and rotational entropy of the ligand upon binding, we observed an additional ca. 1.2 kcal/mol greater entropic loss for the WT compared to the I19G mutant (Fig. S28). Therefore binding of AM-7209 incurs an overall roto/translational and conformational entropy loss that is greater for WT than I19 by ca. 2.2 kcal/mol. These findings are consistent with the differential changes in HetNOE NMR measurements in the lid region between WT and I19G (Fig. 4D) and the ITC measurements that show a ca. 3 kcal/mol less favourable entropy of binding for AM-7209 to WT than for I19G (Fig. 3C).

Taken together these results suggest that formation of a stable helical motif in residues 21–25 at the base of the MDM2 lid region requires additional contacts between the methyl groups of I19, the methyl group of T16 and the chlorophenyl ring of AM-7209 (Fig. 5F). Disruption of this network of hydrophobic contacts increases lid flexibility and unwinds the helical motif. This explanation also accounts for the significant loss of affinity of AM-7209 for the T16G mutant observed in ITC experiments (Fig. 3C). Thus, these results suggest that ordering of the MDM2 lid into a helical turn-extended motif is driven by shielding of a cluster of protein and ligand hydrophobic moieties. This mechanism is reminiscent of hydrophobic collapse observed in protein folding. To support this interpretation additional simulations were carried out with a modified version of AM-7209 in which the chlorophenyl moiety pointing towards T16 was replaced by a phenyl ring (AM-7209-Cl, Fig. S25). The decreased contacts between the ligand and T16 were observed to destabilise the helical motif in a manner qualitatively similar to the effects of the I19G mutation (Fig. 5E, bottom panel). These findings provide an experimentally testable hypothesis for the proposed hydrophobic collapse mechanism.

### Conclusions

This study combined calorimetric and NMR measurements with detailed atomistic molecular simulations of protein–ligand interactions to elucidate the molecular recognition mechanism that underpins ordering of the N-terminal MDM2 lid region upon binding of AM-7209, a small molecule ligand. This ligand-specific lid ordering is found to be driven by the shielding from solvent of a cluster of lid and ligand non-polar moieties. Other lid stabilisation mechanisms, such as direct contacts between lid residue Val14 and AM-7209, or salt bridge formation between lid residue Glu23 and MDM2 residue Arg97, were shown to only play a secondary role. The computer simulations enabled identification of a single residue mutation at Ile19, that abrogates this ligand-induced protein disorder-order transition

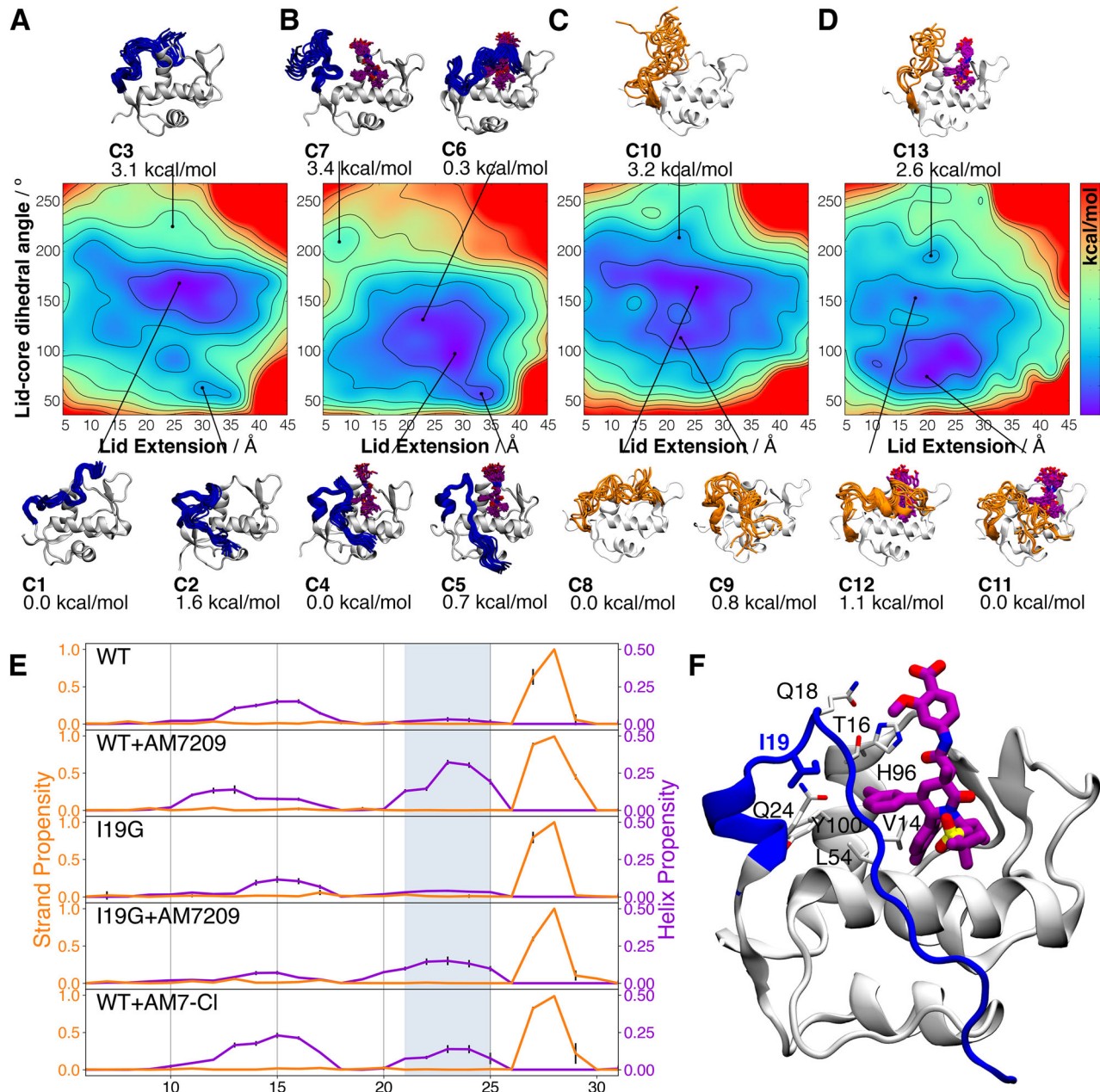

**Fig. 5 | Conformational free energy landscapes of the MDM2 lid IDR. A** Free-energy surface of the lid region of unliganded WT MDM2 obtained along two collective variables that measure the length of the lid ('lid extension') and its position relative to the core region ('lid-core dihedral angle'). See Fig. S20 for additional details. Representative conformations in low free-energy regions are depicted. The free energies of distinct regions are relative to the lowest free-energy region. **B** Same as (**A**) but for WT in complex with AM-7209. **C** Same as (**A**) but for unliganded I19G. **D** Same as (**A**) but for I19G in complex with AM-7209. **E** Conformational preferences of the lid in WT, WT:AM-7209 complex, I19G, I19G:AM-7209 complex and WT:AM-7209-Cl complex. The location of residues involved in the formation of the ordered helical motif is indicated by the shaded blue area. **F** Cartoon showing AM-7209-lid interactions highlighting relative positions of the AM-7209 chlorophenyl moiety and the side-chain of I19 in WT.

and is associated with a concomitant loss of three orders of magnitude in affinity for the ligand. Our NMR and MD data suggest that the I19G mutation does not significantly perturb the conformational ensemble of the MDM2 lid region in its apo form. The I19G mutation disrupts non-polar packing with lid residue Thr16 and the chlorophenyl ring of AM-7209, that are necessary to overcome the conformational entropy loss incurred by ordering the lid into a helix-turn-strand motif. Unlike previous studies highlighting conformational changes of Tyr100 in the presence of diverse ligands[37], AM7209 uniquely stabilises the fully ordered MDM2 lid, with Ile19 identified as a critical driver, as supported by combined theoretical and experimental evidence. Our findings thus suggest that, in addition to loss-of-

function mutations in the TP53 gene, mutations in MDM2's intrinsically disordered lid region have the potential to mediate resistance to some classes of therapeutic p53/MDM2 antagonists[38].

More broadly, our results highlight how holistic consideration of ligand interactions with both folded and disordered protein regions can unlock significant binding affinity and selectivity enhancements. Recent advances have enabled generation of sequence-to-ensemble representations for entire IDR proteomes[3,39]. Given the high prevalence of IDRs, assessment of energy landscapes of IDRs flanking conventional structured protein regions through detailed atomistic simulations may offer a new perspective to exploit cryptic binding sites for small molecules in therapeutic interventions.

## Methods

### Molecular dynamics simulations

The software FESetup[40] was used to prepare inputs for MD simulations of protein–ligand complexes for MDM2 residues 6–125 (WT), its mutants (V14G, V14D, V14T, T16Q, Q18G, Q18E, I19G, I19A, I19V, I19E, E23G, E23L, E23Q, I19G:V14G, V14D:I19G, T16G:I19G, I19G:E23G, I19G:Q24G, V14G:T16G, T16G:Q24G, E23G:Q24G and E23R:R97E) and a truncated version, MDM2 17–125. Simulations were carried out using the SOMD software[41]. Each simulation was performed for 560 ns and analysed with the GROMACS suite[42].

The accelerated molecular dynamics (aMD) method was used to sample the conformational ensembles of the MDM2 lid for each system[43,44]. Bidimensional umbrella sampling (US) was employed to compute the equilibrium distribution of each of the lid conformational ensembles[45]. The 2D variational free energy profile (vFEP) method[46] was used to obtain unbiased free energy profiles along the defined collective variables space. aMD and US simulations were performed using the AMBER16 software package[47] and processed using CPPTRAJ[48]. Entropy changes between conformational ensembles were performed using PDB2ENTROPY and PDB2TRENT[49].

### Protein purification

The gene (uniprot ID = A0A0A8KA17) of MDM2 (residues 6–125 for wild type and mutants) was inserted into a pET20b plasmid (ampicillin-resistant) with a six-his-tag in the C-terminal sequence. Proteins were produced in *Escherichia coli* strains C41 (DE3), C43 and BL21 (DE3), grown in LB broth for non-labelled proteins and minimal medium supplemented with $^{15}NH4Cl$ and/or 13C6-glucose for labelled proteins. Purification was performed using IMAC (immobilised metal ion chromatography-5 mL) followed by size-exclusion chromatography. Proteins were characterised by dynamic light scattering, electrophoresis and LC-MS.

### ITC experiments

ITC was used to measure the dissociation constant ($K_D$) of MDM2 ligands Nutlin-3a and AM-7209. All titrations (10 μM proteins in the cell and 100 μM Nutlin-3a or 150 μM AM-7209 in the syringe) were performed using a MicroCal Auto-iTC200 isothermal titration calorimeter from Malvern Panalytical, assuming one site of binding. The data were analyzed using the MicroCal PEAQ-ITC Analysis Software version 1.1.0.1262.13. For some constructs, measuring the binding affinity directly through titration proved difficult; therefore a competitive titration was used in these cases[50].

### NMR experiments

Chemical shift perturbation measurements and relaxation experiments (T1, T2, heteronuclear ($^1H$,$^{15}N$) NOEs) were performed, in the absence and presence of AM-7209, for WT and I19G (~1 mM and ~480 μM protein concentration, respectively and complexes in molar ratio 1.2:1 ligand:protein). NMRPipe was used to process all the 2D and 3D spectra and subsequently analyzed using NMRFAM-Sparky and POKY[51,52]. The same samples were used for backbone $^1H$, $^{13}C$ and $^{15}N$ chemical shift assignments. Data have been deposited in BMRB (accession numbers 52216, 52219, 52220 and 52221, respectively).

### Reporting summary

Further information on research design is available in the Nature Portfolio Reporting Summary linked to this article.

## Data availability

Detailed simulation and experimental methods. Supplementary Figs. S1–S28; Tables S1–S4 and Datasets S1; S2. All data are available in the main text or the Supplementary Materials. Assigned backbone NMR chemical shifts for WT, WT:AM7209, I19G and I19G:AM7209 have been deposited in the Biological Magnetic Resonance Data Bank (BMRB entry accession numbers 52216, 52219, 52220 and 52221, respectively).

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

## Acknowledgements

Kirsten Ritchie is acknowledged for help preparing and characterising the E23RR97E mutant. Gratitude is expressed to Amgen for providing materials for this research via its extramural research program. Leszek Poppe is acknowledged for sharing assignments of chemical shifts for apo WT MDM2. We would like to acknowledge Juraj Bella from the NMR facility at UoE for his useful guidance during the NMR experiments acquisition. We also want to thank the NMRbox cloud computing platform for access to NMRPipe and other NMR data processing packages.

## Author contributions

Conceptualisation: J.M.; methodology: C.M-M., A.A.G., S.L., P.N.B. and J.M.; investigation: C.M-M., A.A.G. and S.L.; visualisation: C.M-M., A.A.G., S.L. and J.M.; supervision: P.N.B., J.M.; writing—original draft: C.M-M., A.A.G., S.L. and J.M.; writing—review and editing: C.M-M., A.A.G., S.L., P.N.B. and J.M.

## Competing interests

The authors declare no competing interests.
