## [Transparent Peer Review File · Communications Chemistry]

Molecular Driving Force of a Small Molecule-Induced Protein Disorder-Order Transition

Corresponding Author: Professor Julien Michel

Version 0:

Reviewer comments:

Reviewer #1

(Remarks to the Author)

In this work, Mendoza-Martinez et al. investigated the mechanism underlying the folding of the N-terminal intrinsically disordered lid region of MDM2 by the small molecule AM-7209 through a combination of molecular dynamics simulations, calorimetry, and NMR measurements. They identified that a network of non-polar contacts between a chlorophenyl moiety of AM-7209 and the lid residue I19 plays a crucial role in driving the folding of the lid. Intrinsically disordered regions play a crucial role in protein-protein interactions and may serve as potential drug targets for various diseases. However, due to their conformational flexibility, targeting IDRs for drug design is much more challenging than folded proteins. The findings reported in this work are of great significance. The experiment design was rigorous and the analysis was solid. I would suggest the publication of this work after addressing the following concerns.

1. Conformational sampling (Figure 5A-D) suggests that the lid adopts different conformations in the WT and I19G MDM2. A contact map analysis could provide further insights into the interactions between AM-7209 and MDM2, particularly for residue I19.
2. Figure 2A: The labeling of residues is confusing, as some labels are not placed close to the residues.
3. Figure 5E, bottom panel (WT + AM7-Cl): The ligand AM7-Cl is not mentioned in the main text, so the meaning of this panel is unknown.

Reviewer #2

(Remarks to the Author)

Mendoza-Martinez et al. studies the molecular mechanism of the disorder-to-order transition of an IDR driven by the binding of a small molecule. The study uses combination of molecular dynamics simulations, NMR and ITC to test the effect of a series of mutations in the lid domain, and the data is generally of high quality and interpreted appropriately. The main finding of the manuscript revolves around the mutation I19G, which has a large effect on binding of the compound AM-7209. Several lines of evidence suggest that this occurs because the lid fails to fold upon binding, and the mutation thus behaves as a lid truncation mutation. It is not clear that there is a consequence on the function of the proteins, however the mechanism of structuration of an IDR by a small molecule is quite novel and deserves to be studied at a more conceptual level such as here. I have no major concerns with the manuscript and think it is more or less publishable as is.

Reviewer #3

(Remarks to the Author)

Mendoza-Martinez and colleagues reported that the intrinsically disordered region (IDR) of MDM2 is important for the potency of the inhibitor AM-7209, which targets this oncoprotein and impairs its interaction with p53. The authors used a combination of molecular dynamics simulations, calorimetry, and NMR measurements to identify which residues have the greatest impact on this behavior. Among them, I19 seems to be the crucial one.

Overall, while this work is not groundbreaking, it still has value in providing additional insights into the mechanism of action of AM-7209. Therefore, the research might be exploited by others working with inhibitors of proteins that have some IDR in close proximity to the binding sites.

My major concern is that a similar work by Bista et al. (Structure, 2013) was published twelve years ago (!), and the authors

of the current paper did not mention it at all in their manuscript. Therefore, to highlight the novelty of the current work, the authors must address this paper and its findings in their manuscript.

The molecular dynamics simulations correlates well with experimental data (ITC). There are no major flaws in the data analysis, interpretation, and conclusions. The methodology section is written well. The authors are asked to change 1 and 15 in ¹H, ¹⁵N notation to superscripts.

I will be happy to see the revised version of the paper.

Version 1:

Reviewer comments:

Reviewer #3

(Remarks to the Author)

The authors properly addressed my concerns. I recommend the publication of paper as it is.

Mendoza Martinez *et al.*,

Molecular Driving Force of a Small Molecule-Induced Protein Disorder-Order Transition

Responses to Reviewer Comments

Reviewer #1:

In this work, Mendoza-Martinez et al. investigated the mechanism underlying the folding of the N-terminal intrinsically disordered lid region of MDM2 by the small molecule AM-7209 through a combination of molecular dynamics simulations, calorimetry, and NMR measurements. They identified that a network of non-polar contacts between a chlorophenyl moiety of AM-7209 and the lid residue I19 plays a crucial role in driving the folding of the lid. Intrinsically disordered regions play a crucial role in protein-protein interactions and may serve as potential drug targets for various diseases. However, due to their conformational flexibility, targeting IDRs for drug design is much more challenging than folded proteins. The findings reported in this work are of great significance. The experiment design was rigorous and the analysis was solid. I would suggest the publication of this work after addressing the following concerns.

1.1: Conformational sampling (Figure 5A-D) suggests that the lid adopts different conformations in the WT and I19G MDM2. A contact map analysis could provide further insights into the interactions between AM-7209 and MDM2, particularly for residue I19.

R1.1: We have now added a heatmap for both complexes WT:AM7209 and I19G:AM7209. The data shows that some interactions with the lid are changing, especially in the surroundings of I19 upon mutation (P20, E23, Q24, E25, L27). The Figure has been placed in the SI (Fig. S29).

1.2: Figure 2A: The labelling of residues is confusing, as some labels are not placed close to the residues.

R1.2: Figure 2A has now been updated with improved labelling.

1.3: Figure 5E, bottom panel (WT + AM7-Cl): The ligand AM7-Cl is not mentioned in the main text, so the meaning of this panel is unknown.

R1.3: The ligand was mentioned in the main text in the last paragraph of the section 'Free energy calculations identify the structural requirements for ordering of the MDM2 lid by AM-7209'. We have added an explicit reference to the 'bottom panel' of Figure 5E to attract attention to this data.

Reviewer #2:

Mendoza-Martinez et al. studies the molecular mechanism of the disorder-to-order transition of an IDR driven by the binding of a small molecule. The study uses combination of molecular dynamics simulations, NMR and ITC to test the effect of a series of mutations in the lid domain, and the data is generally of high quality and interpreted appropriately. The main finding of the manuscript revolves around the mutation I19G, which has a large effect on binding of the compound AM-7209. Several lines of evidence suggest that this occurs because the lid fails to fold upon binding, and the mutation thus behaves as a lid truncation mutation. It is not clear that there is a consequence on the function of the proteins, however the mechanism of structuration of an IDR by a small molecule is quite novel and deserves to be studied at a more conceptual level such as here. I have no major concerns with the manuscript and think it is more or less publishable as is.

R2.1: We appreciate the recognition of our work. We believe this study may contribute to a deeper understanding of IDRs in drug design.

Reviewer #3:

Mendoza-Martinez and colleagues reported that the intrinsically disordered region (IDR) of MDM2 is important for the potency of the inhibitor AM-7209, which targets this oncoprotein and impairs its interaction with p53. The authors used a combination of molecular dynamics simulations, calorimetry, and NMR measurements to identify which residues have the greatest impact on this behavior. Among them, I19 seems to be the crucial one.

Overall, while this work is not groundbreaking, it still has value in providing additional insights into the mechanism of action of AM-7209. Therefore, the research might be exploited by others working with inhibitors of proteins that have some IDR in close proximity to the binding sites. My major concern is that a similar work by Bista et al. (Structure, 2013) was published twelve years ago (!), and the authors of the current paper did not mention it at all in their manuscript. Therefore, to highlight the novelty of the current work, the authors must address this paper and its findings in their manuscript.

The molecular dynamics simulations correlates well with experimental data (ITC). There are no major flaws in the data analysis, interpretation, and conclusions. The methodology section is written well. The authors are asked to change 1 and 15 in ¹H, ¹⁵N notation to superscripts.

I will be happy to see the revised version of the paper.

R3.1: The paper by Bista et al. focused on conformational changes that define alternative binding modes of MDM2 inhibitors. In their study, Tyr100 was identified as the key residue driving different lid positions, but the ordered state of the lid was neither observed nor discussed, and Ile19 was not considered critical for binding. By contrast, our work demonstrates that binding of AM7209 preserves the ordered state of the N-terminal lid. We provide both theoretical and experimental evidence that many residues along the lid, particularly Ile19, participate in a cooperative hydrophobic network that stabilizes this ordered state. This reveals a novel mechanism of ligand-induced lid stabilization that was not addressed in the previous study and highlights the unique binding mode of AM7209 compared to other inhibitors. We have added the following sentence in the main manuscript: "Unlike previous studies highlighting Tyr100 in diverse binding modes, AM7209 uniquely stabilizes the fully ordered MDM2 lid, with Ile19 identified as a critical driver, as supported by combined theoretical and experimental evidence." Now our work cites Bista et al. work in the main manuscript as well.

Additionally, we have made corrections to the superscripts of all isotopes mentioned in the manuscript.